

# The comparative population genetics of *Neisseria meningitidis* and *Neisseria gonorrhoeae*

Lucile Vigué[1] and Adam Eyre-Walker[2]

[1] Ecole Polytechnique, Paris, France
[2] School of Life Sciences, University of Sussex, Brighton, UK

## ABSTRACT

*Neisseria meningitidis* and *N. gonorrhoeae* are closely related pathogenic bacteria. To compare their population genetics, we compiled a dataset of 1,145 genes found across 20 *N. meningitidis* and 15 *N. gonorrhoeae* genomes. We find that *N. meningitidis* is seven-times more diverse than *N. gonorrhoeae* in their combined core genome. Both species have acquired the majority of their diversity by recombination with divergent strains, however, we find that *N. meningitidis* has acquired more of its diversity by recombination than *N. gonorrhoeae*. We find that linkage disequilibrium (LD) declines rapidly across the genomes of both species. Several observations suggest that *N. meningitidis* has a higher effective population size than *N. gonorrhoeae*; it is more diverse, the ratio of non-synonymous to synonymous polymorphism is lower, and LD declines more rapidly to a lower asymptote in *N. meningitidis*. The two species share a modest amount of variation, half of which seems to have been acquired by lateral gene transfer and half from their common ancestor. We investigate whether diversity varies across the genome of each species and find that it does. Much of this variation is due to different levels of lateral gene transfer. However, we also find some evidence that the effective population size varies across the genome. We test for adaptive evolution in the core genome using a McDonald–Kreitman test and by considering the diversity around non-synonymous sites that are fixed for different alleles in the two species. We find some evidence for adaptive evolution using both approaches.

## INTRODUCTION

The two closely related bacteria *Neisseria meningitidis* and *N. gonorhoeae* are major human pathogens. *N. gonorhoeae* is the causative agent of the sexually transmitted disease gonorrhoeae which currently infects 106 million people each year worldwide (*WHO, 2012*). When untreated, gonoccocal infections can result in long-term problems such as persistent urethritis, cervicitis, proctitis, pelvic inflammatory disease, infertility, first-trimester abortion, ectopic pregnancy and maternal death (*WHO, 2012*). They also increase the risk of acquiring and transmitting HIV. In cases of pregnancy, *N. gonorhoeae* infections can cause severe damages to neonatal health (*WHO, 2012*).

Corresponding author
Adam Eyre-Walker,
a.c.eyre-walker@sussex.ac.uk

In contrast, *N. meningitidis* is a human commensal infecting approximately 10% of the healthy human population (*Claus et al., 2005*; *Yazdankhah et al., 2004*), which only occasionally causes disease. However, it can cause meningococcal meningitidis and septicaemia with mortality rates that can reach 50% when untreated, and the global disease burden is estimated to be ~500,000 cases a year (*Roberts, 2008*). Among the different micro-organisms that can cause meningitidis, it is regarded as one of the most important because of its ability to cause large epidemics.

Here, we consider several aspects of the population genetics of these bacterial species. The two species are sister taxa (*Bennett et al., 2012*), and *N. meningitidis* is known to be considerably more diverse than *N. gonorrhoeae* within the genes that they share in common (*Bennett et al., 2007*, *2012*). The first problem we address is why the two taxa differ in their diversities. There are several potential explanations. First, *N. gonorrhoeae* might have a lower effective population size, either because it evolved from *N. meningitidis* and went through a bottleneck when the species was formed (*Vazquez et al., 1993*), or because it generally has a lower effective population size, possibly because it has a lower census population size. Second, *N. gonorrhoeae* might have a lower mutation rate than *N. meningitidis*. Third, *N. gonorrhoeae* might acquire less diversity through recombination than *N. meningitidis*. Both *N. gonorrhoeae* and *N. meningitidis* are known to be naturally transformable, and it has been known for many years that both species acquire diversity, within their core genome, by homologous recombination with genetically divergent strains (*Spratt, 1988*; *Spratt et al., 1989*). We refer to this process as homologous lateral gene transfer (hLGT), to differentiate it from the acquisition of accessory genes by non-homologous lateral gene transfer (nhLGT) (however, note that the acquisition of new genes generally involves homologous recombination with flanking genes, so nhLGT will typically involve some hLGT; *Kong et al., 2013*). hLGT leads to mosaic genes, in which parts of the gene have been acquired from a highly divergent strain or a different bacterial species. In fact, *N. meningitidis* and *N. gonorrhoeae* were some of the first bacteria in which this form of recombination was demonstrated (*Spratt, 1988*; *Spratt et al., 1989*). It has been estimated that *N. meningitidis* acquires single nucleotide polymorphisms (SNPs) through hLGT at a rate between 4 and 100× higher than via mutation (*Feil et al., 2001*; *Hao et al., 2011*; *Kong et al., 2013*; *Vos & Didelot, 2009*). In contrast this ratio has recently been estimated to be only about twofold in *N. gonorrhoeae* (*Ezewudo et al., 2015*). It is unclear whether these ratios are significantly different. We investigate this here.

The second question we address is whether diversity varies across the core genome of the two species. Genetic diversity is known to vary across the genome of many species. This was originally demonstrated in *Drosophila melanogaster* by *Begun & Aquadro (1992)* who showed that diversity was positively correlated to the rate of recombination. This was thought to be due to the effects of linked selection, in the form of genetic hitch-hiking (*Smith & Haigh, 1974*) and background selection (*Charlesworth, Morgan & Charlesworth, 1993*), depressing diversity in regions of the genome with low rates of recombination. Variation in diversity across the genome has been demonstrated in many other species including the bacterium *Escherichia coli* (*Maddamsetti et al., 2015*;

*Martincorena, Seshasayee & Luscombe, 2012*). The reasons for this variation remain unclear (*Chen & Zhang, 2013*; *Maddamsetti et al., 2015*; *Martincorena & Luscombe, 2013*).

The final question we address is whether *N. meningitidis* and *N. gonorrhoeae* have undergone adaptive evolution. *N. meningitidis* and *N. gonorrhoeae* inhabit different niches and one presumes they have undergone adaptive evolution to allow them to do this. Some of this adaptation may have been through the acquisition of new genes via nhLGT, but there might also be adaptation in the core genome. Two recent analyses using the $d_N/d_S$ test on the core genome have found limited evidence for adaptive evolution in *N. meningitidis* (*Yu et al., 2014*) and *N. gonorrhoeae* (*Ezewudo et al., 2015*), but this test is known to be very conservative. Here, we apply two additional tests.

## MATERIALS AND METHODS

### Dataset

All 15 genomes of *N. gonorrhoeae* that were present in Genbank in April 2018 (NCCP11945 (*Chung et al., 2008*), FA19 (*Abrams, Trees & Nicholas, 2015*), FA6140 (*Abrams, Trees & Nicholas, 2015*), 35/02 (*Abrams, Trees & Nicholas, 2015*), FA1090, MS11, FA19, FA6140, 35/02, 32867, 34530, 34769, FDAARGOS 204, FDAARGOS 205, FDAARGOS 207, NCTC13799, NCTC13798, NCTC13800) and 20 randomly selected genomes of *N. meningitidis* (MC58 (*Tettelin et al., 2000*), Z2491 (*Parkhill et al., 2000*), FAM18 (*Bentley et al., 2007*), 053442 (*Peng et al., 2008*), alpha14 (*Schoen et al., 2008*), 8013 (*Rusniok et al., 2009*), alpha710 (*Joseph et al., 2010*), WUE 2594 (*Schoen et al., 2011*), G2136 (*Budroni et al., 2011*), M01-240149 (*Budroni et al., 2011*), M04-240196 (*Budroni et al., 2011*), H44/76 (*Budroni et al., 2011*), M01-240355 (*Budroni et al., 2011*), NZ-05/33 (*Budroni et al., 2011*), 510612 (*Zhang et al., 2014*), NM3686, M7124, NM3682, NM3683, L91543) were downloaded from Genbank. From these all protein coding sequences were extracted. We retained those coding sequences that started NTG, terminated with TAA, TAG or TGA and had a length that is a multiple of three. We identified orthologs using reciprocal BLAST, with an e-value threshold of 0.00001; i.e. each protein coding gene in each genome was BLASTed against the genes of FA1090, and then the best hit was BLASTed back onto the original genome, retaining only those hits in which the original query sequence was the best hit. Similar selections of genes were obtained using alternative starting genomes. The protein sequences of the orthologs were aligned using MUSCLE (*Edgar, 2004*). We selected genes where the alignments meet these criteria: the number of gaps is lower than 1% of the length of the sequence and the total number of nucleotides in gaps is lower than 10% of the total number of nucleotides in the sequence. Sequences with internal stop codons were removed. This resulted in a dataset of 1,145 genes belonging to the core genome of both *N. gonorrhoeae* and *N. meningitidis*. We used the BioPython Phylo library (*Cock et al., 2009*) to estimate a phylogeny of the strains based on the core genome alignment.

### Analyses

In most analyses we treated genes independently. However, to detect hLGT we ran ClonalFrameML (*Didelot & Wilson, 2015*) on a concatenation of the protein coding sequences from the core genome of both species. Genes were concatenated randomly

without respect for synteny. For some analyses we masked those regions inferred to be due to hLGT in the strains affected.

We investigated whether linkage disequilibrium (LD) declines with the distance between sites by measuring the LD between all pairs of polymorphisms within each gene; we did not concatenate the genes or align whole genomes, because with the gain and loss of genes the distance between sites differs depending on the strains being analysed. We measured LD using the $r^2$ statistic (*Hill & Robertson, 1968*). LD values were then assigned to bins based on the distance between the two sites—10 bp bins between 1–100 bp, a bin from 101–200 bp and then 200 bp bins between 201–800 bp. We took the average LD and distance between sites for each bin in a manner which weighted each gene equally—we estimated the average LD and distance for pairs of sites in each bin for each gene and then averaged those values across genes. To estimate the approximate half-life of LD, we found the distance between sites that gave approximately half the LD between the LD for the 1–10 bp bin and the asymptotic value of the LD.

Because $r^2$ is constrained to be positive, the expected value of $r^2$ is greater than zero even when there is no LD. To calculate the expected value of $r^2$ when there is no LD, we considered two bi-allelic loci with alleles at frequencies $p_1$ and $p_2$. The expected frequencies of the four haplotypes are $p_1p_2$, $p_1(1-p_2)$... etc. from which we generated four random variates from a multinomial distribution for a sample size of $N$ chromosomes using Mathematica version 11; for each sample of haplotypes we calculated $r^2$. We repeated this procedure 10,000 times and calculated the mean to estimate the expected value of $r^2$. We found that the expected value of $r^2$ is independent of the allele frequencies.

To investigate the relationship between the non-synonymous, $\pi_N$, and synonymous, $\pi_S$, nucleotide diversity we used a variation of the method of *James, Castellano & Eyre-Walker (2017)* to combine data from different genes. If the distribution of fitness effects of new mutations is a gamma distribution (assuming most mutations are deleterious) then $\log(\pi_N)$ is expected to be linearly correlated to $\log(\pi_S)$ if there is variation in $N_e$ (*Welch, Eyre-Walker & Waxman, 2008*). However, for many genes either $\pi_N$ or $\pi_S$ is zero, hence we need to combine genes together. We can do this by splitting the synonymous polymorphisms into two groups according to whether they were in an odd or even numbered codon and then using the two groups to estimate two synonymous nucleotide diversities that have independent sampling errors, $\pi_{S1}$ and $\pi_{S2}$. One of these, $\pi_{S1}$, was used to rank and group genes, and the other was averaged across genes in the group to give an unbiased estimate of $\pi_S$ for the group. $\pi_N$ was also averaged across the genes in the group.

To investigate the diversity around sites that are fixed between *N. meningitidis* and *N. gonorrhoeae* for different alleles we focused on genes that had at least one synonymous polymorphism and one fixed difference between the two species. For each fixed difference, we identified all the synonymous polymorphisms that were within one kb and we grouped them by windows of 100 bp. Since, background selection can potentially lead to a lower dip in diversity around fixed non-synonymous mutations we normalised the diversity around fixed synonymous and non-synonymous substitutions by dividing the number of synonymous polymorphisms in a particular window by the total number of synonymous polymorphisms in the gene, multiplied by the window size over the gene length.

**Table 1 Nucleotide diversity estimates across all sites in the core genome (pi) and at zerofold non-synonymous sites (pi$_N$) and fourfold synonymous sites (pi$_S$).**

|  | π | π$_S$ | π$_N$ |
|---|---|---|---|
| *N. gonorrhoeae* | 0.0029 (0.0008) | 0.007 (0.002) | 0.0014 (0.0004) |
| *N. meningitidis* | 0.022 (0.007) | 0.06 (0.02) | 0.007 (0.002) |
| Ratio | 7.6 | 8.6 | 5.0 |

**Table 2 Recombination rate estimates obtained from ClonalFrameML along with their 95% confidence intervals.**

| Species | *R/θ* | δ | *v* (%) | *r/m* | θ (× 10⁻³) | *R* (× 10⁻⁴) |
|---|---|---|---|---|---|---|
| *N. gonorrhoeae* | 0.41 (0.39, 0.43) | 70 (67, 72) | 6.9 (6.8, 7.1) | 2.0 (1.8, 2.2) | 1.0 (0.8, 1.2) | 4.0 (3.0, 5.0) |
| *N. meningitidis* | 1.2 (1.2, 1.3) | 99 (98, 100) | 5.3 (5.3, 5.4) | 6.4 (6.2, 6.7) | 3.0 (2.5, 3.5) | 36 (30, 44) |
| Ratio (*N. meningitidis*/*N. gonorrhoeae*) | 3.0 | 1.4 | 0.77 | 3.2 | 3.0 | 9.3 |

**Note:**

Given is the rate at which recombination tracts initiate (*R*) relative to the rate of mutation (theta), both multiplied by the effective population size, the average length of recombination tracts (delta) and the proportion of sites that differ to the resident sequence (mu), along with the rate at which sites change due to recombination relative to mutation (*r/m*).

# RESULTS

## Recombination and mutation

We are interested in how genetic variation is generated and distributed in the two *Neisseria* species *N. meningitidis* and *N. gonorrhoeae*. Although, the presence and absence of genes in the strains of the two species is an important aspect of this problem, here we focus on the genetic variation that is present in the core genome that is common to both species. Using reciprocal BLAST, we identified 1,145 genes present across the 15 genomes of *N. gonorrhoeae* and 20 genomes of *N. meningitidis* that we analysed. The total length of this core genome is 1.1 MB long. Defining a polymorphism as a site that contains two or more alleles within either of the two species, we find that *N. meningitidis* is ~7.6-fold more diverse than *N. gonorrhoeae* consistent with previous qualitative reports (*Bennett et al., 2007, 2012*). The difference in diversity is more apparent at synonymous (~8.9-fold) than non-synonymous (~5.5-fold) sites (Table 1), a pattern we return to later. The two species share a modest amount of diversity; 35% of all polymorphisms in *N. gonorrhoeae* are shared with *N. meningitidis*, and 4.5% of those in *N. meningitidis* are shared with *N. gonorrhoeae*.

It is well known that *N. meningitidis* and *N. gonorrhoeae* undergo substantial levels of homologous recombination with divergent strains, possibly from other species of bacteria. This leads both to the acquisition of new genes, but also to the acquisition of parts of genes that are already present in the genome; we refer to these processes as nhLGT and hLGT, respectively. To quantify the role that hLGT plays in the acquisition of diversity in the core genome we ran ClonalFrameML (*Didelot & Wilson, 2015*). The method estimates the ratio of the rate at which recombination tracts initiate (*R*) and the rate of mutation (θ), both multiplied by twice the effective population size, $N_e$, along with the average recombination tract length, δ, and the proportion of sites that differ between the imported and resident sequences, *v*. Estimates of these parameters are given in Table 2.

The overall effect of recombination relative to mutation can be estimated as $R\delta\nu/\theta = r/m$, where $r$ and $m$ are the rates at which variants are introduced into a genome by recombination and mutation, respectively.

In *N. meningitidis* we find that recombination introduces 6.43 (95% CI [6.16–6.71]) times more variation than mutation, whereas in *N. gonorrhoeae* it introduces 1.97 (1.76–2.19) times as much. In *N. meningitidis* the $r/m$ ratio has previously been estimated to be 5.37 (*Hao et al., 2011*), 6.71 (*Vos & Didelot, 2009*), 16.4 (*Kong et al., 2013*) and 100 (*Feil et al., 2001*). Our estimate is similar to the first two estimates, but substantially lower than the last two estimates. Both of these latter estimates were obtained from very closely related strains and hence may reflect the value of $r/m$ before natural selection has had an opportunity to operate. In *N. gonorrhoeae* it has been estimated that 2.2× as much variation is introduced by recombination (*Ezewudo et al., 2015*), which is very similar to our estimate. The estimates of $r/m$ mean that ~87% of all polymorphisms in *N. meningitidis* are a consequence of recombination, whereas in *N. gonorrhoeae* it is 66%. The difference between the two species in the influence of recombination is largely driven by a difference in the ratio of the rate at which recombination is initiated vs. the mutation rate ($R/\theta$), since although the tract lengths are estimated to be on average slightly longer in *N. meningitidis*, they introduce slightly less variation than *N. gonorrhoeae* (Table 2).

ClonalframeML estimates the ratio of $R$ and $\theta$ but not their absolute values. However, we can estimate the absolute value as follows. We note that the nucleotide diversity is due to the input of mutation and the input of recombination: i.e. $\pi = \theta + R\delta\nu$. If we note that ClonalframeML gives us an estimate of $R/\theta$ we can rewrite this equation as $\pi = \theta + \theta\delta\nu R/\theta$, from which we can estimate $\theta = \pi/(1 + \delta\nu R/\theta)$. Estimates of $R$ and $\theta$ are given in Table 2. From this it is evident that the nucleotide diversity is higher in *N. meningitidis* both because of a threefold greater mutational input and a ninefold greater rate at which recombination tracts initiate in *N. meningitidis*, at the population level (i.e. when the tract length initiation rate and mutation rate are multiplied by $N_e$).

The parameters $R$ and $\theta$ are the rates of recombination initiation and mutation, multiplied by the effective population size. Hence, a simple reason why both parameters are higher in *N. meningitidis* might simply be that *N. meningitidis* has a higher $N_e$ than *N. gonorrhoeae*. To test this idea, we masked all sequences that were identified as due to hLGT by ClonalframeML and estimated the levels of non-synonymous and synonymous diversity. Under a model in which synonymous mutations are neutral and non-synonymous mutations are deleterious, but drawn from some distribution, we expect $\pi_N/\pi_S$ to be lower in species with high $N_e$; this is because selection is more effective in species with higher $N_e$ and hence the proportion of mutations that are effectively neutral is lower (*Ohta, 1972*, *1977*, *1992*). This is what we find—$\pi_N/\pi_S = 0.095$ (SE = 0.0023) in *N. meningitidis* vs. 0.23 (0.014) in *N. gonorrhoeae*. These are significantly different to each other (normal test $z = 9.5$, $p < 0.001$).

As we described above, *N. meningitidis* and *N. gonorrhoeae* share a modest amount of genetic variation. It is of some interest whether this is a consequence of hLGT or the inheritance of genetic variation from their common ancestor. If we exclude those

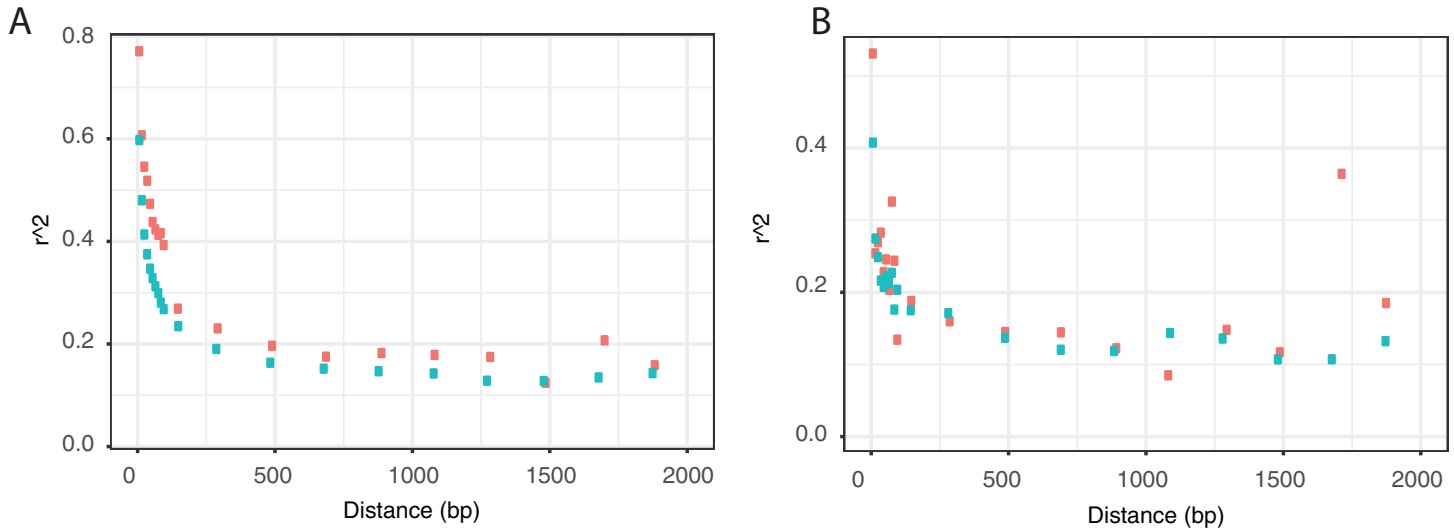

**Figure 1 Decay in linkage disequilibrium with the distance between sites.** Linkage disequilibrium, as measured by $r^2$ (*Hill & Robertson, 1968*), between pairs of polymorphic sites as a function of the distance between sites for (A) all sites and (B) for those sites not inferred to have undergone hLGT. Each point represents the average $r^2$ between all pairs of points separated by a certain distance in bins of 10 bp between 0 and 100 bp, a bin of 101–200 bp and then bins of 200 bp up to 800 bp. *N. meningitidis* in green, *N. gonorrhoeae* in red.

sequences inferred to be due to hLGT we find that the two species still share a modest amount of genetic variation −15.5% of all *N. gonorrhoeae* polymorphisms are shared with *N. meningitidis* and 2.4% of *N. meningitidis* polymorphisms are shared with *N. gonorrhoeae*, approximately half of all shared polymorphisms in each case, suggesting that some proportion of the shared variation originated from their common ancestor.

## Linkage disequilibrium

Homologous recombination can both increase and decrease LD; homologous recombination with divergent strains, of the sort detected by ClonalFrameML, generates LD because it simultaneously introduces many polymorphisms that are initially linked to each other. However, homologous recombination amongst a set of closely related strains breaks-up LD. To investigate how these two forces play out, we calculated the LD between all pairs of sites within each gene and plotted these as a function of the distance between sites. As expected, we observe a decline in LD with distance (Fig. 1A). Both species show similar patterns with LD declining rapidly; in *N. meningitidis* the approximate half-life is 30 bp and in *N. gonorrhoeae* it is 100 bp. The decline could be due to two processes. If most hLGT fragments tend to be short, with decreasing numbers of long fragments, then LD will be greater between closely linked sites. However, we also expect a decline due to recombination between closely related strains, and in fact we observe a decline even when we focus on those parts of the genome which do not appear to have undergone hLGT (Fig. 1B).

In both species LD asymptotes above zero. The non-zero asymptote could be due to one of three reasons—statistical bias, population substructure and a balance between genetic drift and recombination. The statistical bias arises because our measure of LD,
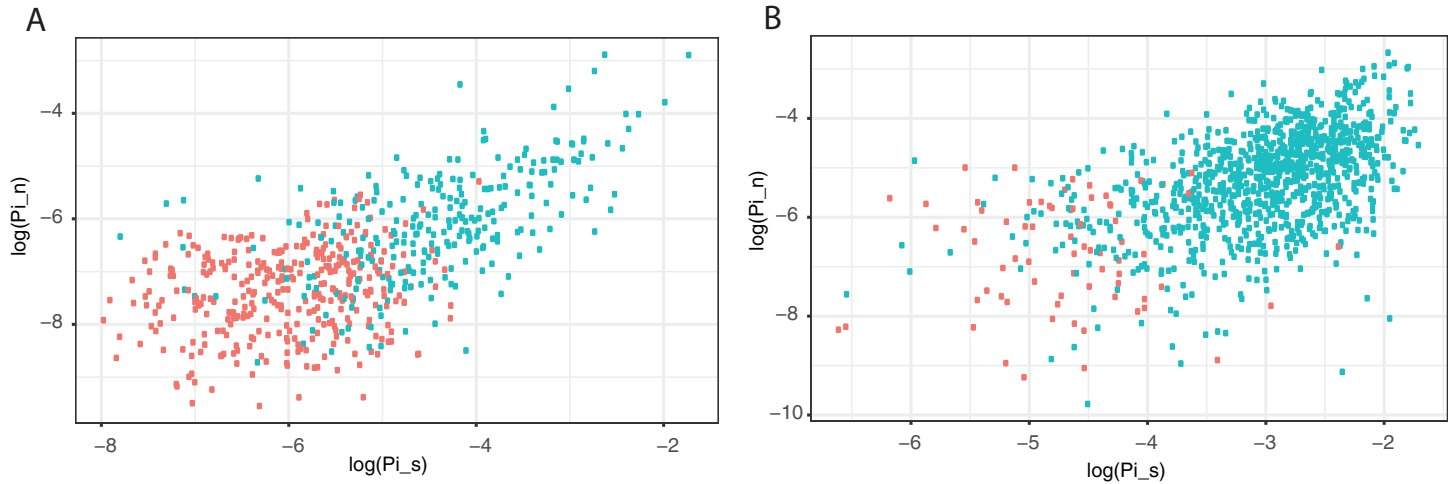

**Figure 2 Correlation between non-synonymous and synonymous diversity across the genome.** The correlation between the log of the non-synonymous nucleotide diversity and the log of the synonymous diversity for core genes in (A) *N. meningitidis* and (B) *N. gonorrhoeae*. Points in green are genes with evidence of hLGT and red are those genes without evidence of hLGT. Note that some genes are excluded because they have either no non-synonymous or synonymous diversity.

$r^2$ (*Hill & Robertson, 1968*), cannot be negative, so positive values of $r^2$ are expected even if there is no LD if sample sizes are small; for sample sizes of 15 and 20 strains, the expected value of $r^2$ is 0.079 and 0.050, respectively (see Materials and Methods), so the asymptote is clearly above this level. Both, *N. meningitidis* and *N. gonorrhoeae* have been shown to have some level of population structure so this is the likely to be part of the explanation (*Budroni et al., 2011*; *Joseph et al., 2011*). However, the slower decay in LD, and higher asymptote in *N. gonorrhoeae*, is consistent with *N. gonorrhoeae* having a smaller $N_e$ than *N. meningitidis*—i.e. the non-zero asymptote might in part be caused by a balance between genetic drift creating LD, and recombination breaking it down.

## Diversity across the genome

Nucleotide diversity is known to vary across the genomes of many organisms. This is largely thought to be driven by variation in the mutation rate or variation in the effects of linked selection. However, in bacteria, and particularly *N. meningitidis* and *N. gonorrhoeae*, it could also be due to variation in the frequency of hLGT. All of these processes are expected to affect synonymous and non-synonymous diversity to greater or lesser extents, and indeed we observe a positive correlation between non-synonymous and synonymous diversity, demonstrating that both vary across the genome in concert. At least part of this pattern is driven by hLGT because genes with hLGT show higher $\pi_N$ and $\pi_S$ values than genes without any evidence of hLGT (Fig. 2).

However, to investigate whether there is also variation in the effective population size across the genome we removed sequences inferred to be due to hLGT by ClonalFrameML from our data. This reduces our data substantially and so to reduce statistical sampling issues we used the method of *James, Castellano & Eyre-Walker (2017)* to combine data from different genes. We find that $\pi_N$ and $\pi_S$ are still significantly correlated suggesting the

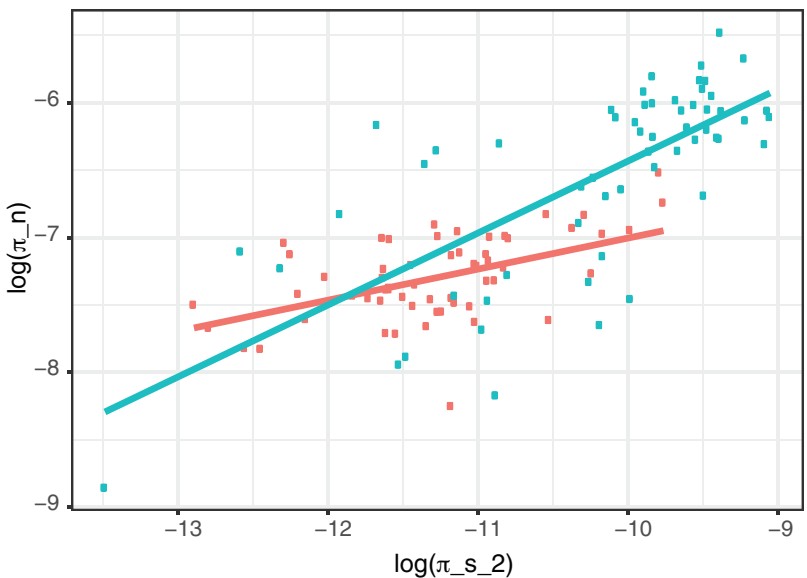

**Figure 3 Correlation between non-synonymous and synonymous diversity excluding regions with evidence of hLGT.** The correlation between the log of the non-synonymous nucleotide diversity plotted and the log of the synonymous diversity for regions of the genome that have not undergone hLGT. Green is *N. meningitidis*, red is *N. gonorrhoeae*. Also shown are the lines of best fit.

correlation between them is not just driven by hLGT (*N. gonorrhoeae* slope = 0.23, $p < 0.001$; *N. meningitidis* slope = 0.53, $p < 0.001$) (Fig. 3). The remaining correlation could be due to variation in the mutation rate or variation in the effects of linked selection. We can test whether there is variation in the effects of linked selection by considering the slope between $\log(\pi_N)$ and $\log(\pi_S)$. Under a model in which there is no variation in linked selection then the slope of this relationship is expected to be one, and if there is variation in linked selection the slope if expected to be less than one (*Galtier, 2016*; *Welch, Eyre-Walker & Waxman, 2008*). Linked selection has two consequences. First, it increases the stochasticity in allele frequencies. For example, the spread of an advantageous mutation or the elimination of deleterious genetic variation, removes linked genetic diversity; whether a linked mutation survives either process is a random process depending on whether the advantageous or deleterious mutation occurs in linkage with the target mutation. This can be thought of as reduction in the effective population size. Second, genetic hitch-hiking leads to non-equilibrium dynamics. After a selective sweep, genetic diversity will recover, but this happens faster for deleterious than neutral mutations (*Brandvain & Wright, 2016*; *Do et al., 2015*; *Gordo & Dionisio, 2005*). In both cases we expect a negative correlation between $\pi_N/\pi_S$ and $\pi_S$, which manifests itself in a positive correlation between $\log(\pi_N)$ and $\log(\pi_S)$ but with a slope of less than one (*James, Castellano & Eyre-Walker, 2017*). We find that the slope of the relationship between $\log(\pi_N)$ and $\log(\pi_S)$ is 0.23 (SE = 0.052) and 0.59 (0.070) for *N. gonorrhoeae* and *N. meningitidis*, respectively, in both cases significantly less than one ($p < 0.001$); i.e. $\pi_N$ increases as $\pi_S$ increases but not as fast. The slopes are significantly different to each other (*t*-test, $p < 0.001$).

## Adaptive evolution

*N. meningitidis* and *N. gonorrhoeae* are ecologically quite different and one presumes the two species have undergone adaptation to live in their respective environments. Some of this adaptation will have come about through the acquisition of whole genes through nhLGT. However, some of the adaptation may have occurred within the core genome of the two species either by new mutations, standing genetic variation, or hLGT. To investigate whether there has been adaptation in the core genome we used two approaches. First, we used the *McDonald & Kreitman (1991)* approach to estimate the rate of adaptive evolution (*Eyre-Walker, 2006*; *Fay, Wycoff & Wu, 2001*). In this method the numbers of non-synonymous and synonymous substitutions (i.e. differences between the two species, $d_N$ and $d_S$, respectively) are compared to the numbers of non-synonymous and synonymous polymorphisms ($p_N$ and $p_S$, respectively). Under a neutral model in which mutations are either neutral or strongly deleterious we expect $d_N/d_S = p_N/p_S$ (*McDonald & Kreitman, 1991*). In contrast if there are slightly deleterious non-synonymous mutations we expect $d_N/d_S < p_N/p_S$, and if there are some advantageous mutations we expect $d_N/d_S > p_N/p_S$ (*Eyre-Walker, 2006*; *Fay, Wycoff & Wu, 2001*). Summing $d_N$, $d_S$, $p_N$ and $p_S$ we calculate the fixation index FI = $d_N p_S/d_S p_N$ (*Gojobori et al., 2007*); adaptive evolution is indicated if FI >1.

We find that our estimate of FI differs if we use the polymorphism data of *N. meningitidis* or *N. gonorrhoeae*; using the SNP data of *N. meningitidis* we estimate that FI is significantly greater than one suggesting adaptive evolution has occurred (FI = 1.51 with 95% Cis = 1.41 and 1.61), but if we use the SNP data of *N. gonorrhoeae*, our estimate is significantly less than one (FI = 0.92 (0.83, 0.99)). Estimates less than one can occur if there are slightly deleterious mutations (SDMs) segregating, but even if we restrict our analysis to common polymorphisms, which should remove many of the SDMs (*Charlesworth & Eyre-Walker, 2008*; *Fay, Wycoff & Wu, 2001*), we find that the FI <1 using the SNP data of *N. gonorrhoeae* (using SNPs with allele frequencies above 15%, FI = 0.78 (0.78, 0.88)). An explanation for why FI differs between the two species is that either *N. meningitidis* has undergone population expansion, or *N. gonorrhoeae* has undergone contraction. If there are SDMs then population size expansion leads to an overestimate of FI whereas contraction leads to an underestimate (*Eyre-Walker, 2002*; *McDonald & Kreitman, 1991*). As we argue above, a simple explanation for why *N. meningitidis* is more diverse than *N. gonorrhoeae* is that *N. meningitidis* has a higher $N_e$. We find no evidence of expansion or contraction amongst the current strains—Tajima's *D* (*Tajima, 1989*), a measure of a skew in the site frequency spectrum away from what we expect for neutral mutations in a stationary population size is close to zero and not significantly different to zero in both species in the regions of the genome that have no evidence of hLGT (Tajima's *D* = −0.073 and −0.093 in *N. meningitidis* and *N. gonorrhoeae*, respectively), consistent with previous analyses in *N. meningitidis* (*Joseph et al., 2011*). However, the expansion or contraction in either *N. meningitidis* or *N. gonorrhoeae* could have occurred sometime in the past which would not be visible to an analysis using Tajima's *D*, but which might still affect the FI.
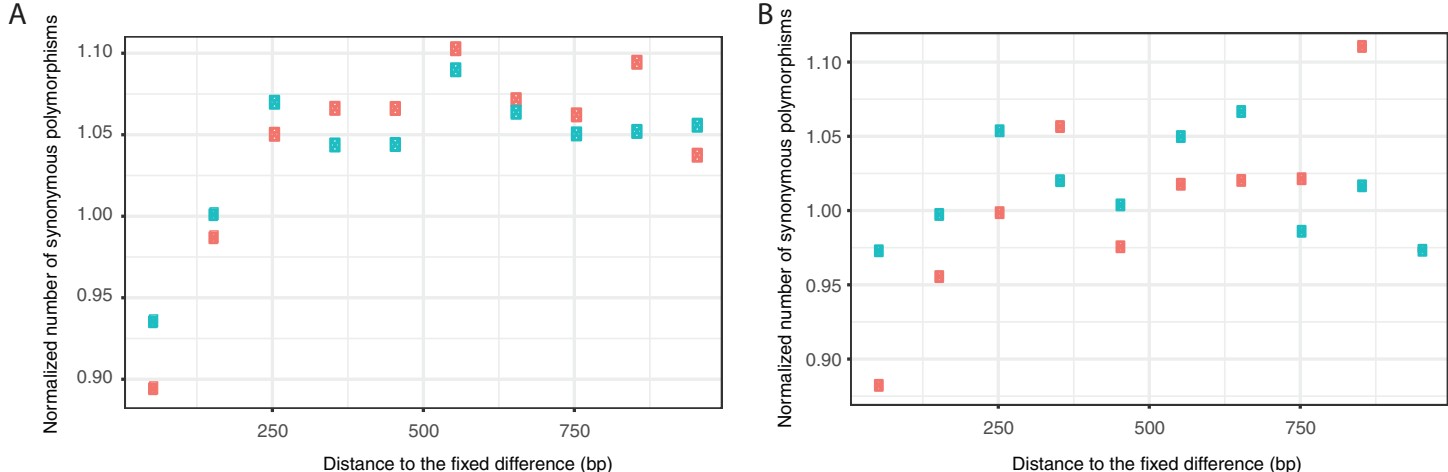

**Figure 4 Synonymous diversity around sites fixed for either non-synonymous or synonymous substitutions.** Average synonymous diversity in (A) *N. meningitidis* and (B) *N. gonorrhoeae* around sites that are fixed for either a non-synonymous (red) or synonymous (green) substitution between *N. meningitidis* and *N. gonorrhoeae*.

A second approach to test for adaptive evolution, is to investigate whether there is a dip in genetic diversity around putatively advantageous mutations (*Sattath et al., 2011*)—as advantageous mutations spread through a population it reduces diversity in its proximity. We find that synonymous diversity is lower close to sites that are fixed for different nucleotides in the two species, and this dip is significantly greater for non-synonymous than synonymous fixed differences when considering diversity in *N. meningitidis* ($p < 0.001$ for distances 1–100, 101–200 and 201–300 bp), consistent with a proportion of non-synonymous mutations being fixed by positive adaptive evolution; a similar pattern is not evident in *N. gonorrhoeae*, possibly because it is less diverse.

There is, however, an alternative explanation for the greater dip around non-synonymous substitutions; if the strength of background selection (*Charlesworth, Charlesworth & Morgan, 1995*) varies across the genome, then regions with high levels of background selection will have low diversity but will tend to also fix slightly deleterious non-synonymous mutations. To investigate whether there is evidence of this, we considered whether the ratio of the non-synonymous to synonymous substitution rates, $\log(d_N/d_S)$, was correlated to synonymous diversity, $\log(\pi_S)$. We again use the method of *James, Castellano & Eyre-Walker (2017)* to combine data from different genes and find that a strong negative correlation in *N. meningitidis* (slope = −0.10, $p = 0.018$) but not in *N. gonorrhoeae* (slope = −0.001, $p = 0.97$). This suggests that background selection might be a factor in *N. meningitidis*. To take into account the potential variation in background selection, we normalised the data from each gene by dividing the number of synonymous SNPs in each window by the average diversity in each gene. This will account for variation in background selection at a gene level, but not at a sub-gene level. The normalised data show a greater dip in diversity for fixed non-synonymous than synonymous substitutions in both *N. meningitidis* (combining *t*-test results from the three closest points, $p < 0.001$), and *N. gonorrhoeae* ($p = 0.0024$) (Fig. 4) although the differences are not large.

## DISCUSSION

We have investigated several aspects of the comparative population genetics of the two bacteria *N. meningitidis* and *N. gonorrhoeae*. We find, as others have (*Bennett et al., 2007*, *2012*), that *N. meningitidis* is substantially more diverse than *N. gonorrhoeae*, but that the two species share a moderate amount of diversity in the genes that they have in common. This shared diversity could have been a consequence of ancestral polymorphism that has been inherited by both species, or due to hLGT transferring variation between the two. We find a substantial fraction is indeed due to hLGT, since if we remove the fraction of the genome that appears to have undergone hLGT, the fraction of shared polymorphism drops considerably. However, there is some diversity that appears to have been inherited from the ancestor.

In both species we find that most of their genetic diversity has been acquired by recombination, rather than by mutation. In *N. meningitidis* we estimate that the total input from hLGT is sixfold greater than from mutation; this is in line with the estimates of *Hao et al. (2011)* and *Vos & Didelot (2009)*, but lower than two other estimates (*Feil et al., 2001*; *Kong et al., 2013*). Both of these high estimates were derived by considering very closely related strains. If hLGT events are on average more deleterious than single nucleotide changes then we expect *r/m* estimates to be greater for more closely related strains, because natural selection has had more opportunity to remove the deleterious mutations in distantly related strains. This has the implication that *r/m* may be far higher amongst newly arising mutations than often thought. In *N. gonorrhoeae* we find the input of hLGT is twofold greater than mutation, consistent with the one previous estimate performed on a similar selection of strains (*Ezewudo et al., 2015*).

*N. meningitidis* might be more diverse than *N. gonorrhoeae* either *N. meningitidis* has a higher mutation rate, a greater rate of hLGT or a higher effective population size. Several lines of evidence suggest that *N. meningitidis* has a higher $N_e$. First, *N. meningitidis* has higher values of both $R$ and $\theta$, where $R$ and $\theta$ are estimates of the rate at which recombination initiates and the mutation rate, multiplied by $N_e$. Second, $p_N/p_S$ is lower in *N. meningitidis* in the fraction of the genome which does not seem to have undergone hLGT. Third, LD declines faster in *N. meningitidis* and asymptotes at a lower level. However, this does not preclude a role for either faster rates of mutation or recombination in the greater diversity in *N. meningitidis*.

It is possible that the lower $N_e$ in *N. gonorrhoeae* is due to a bottleneck at the time when *N. gonorrhoeae* was formed, assuming that it is a derivative of *N. meningitidis* (*Vazquez et al., 1993*). Alternatively, it may be due to the fact that *N. gonorrhoeae* has a lower census population size. Currently ~10% of the human population is asymptomatically infected with *N. meningitidis* (*Claus et al., 2005*; *Yazdankhah et al., 2004*), whereas levels of *N. gonorrhoeae* infection are thought to be very low—between 1 and 170 cases per 100,000 individuals in Western Europe and America in 2017 (www.cdc.gov, ecdec.europa.eu). Hence, although there seems to be a poor correlation between census and effective population size across species (*Bazin, Glemin & Galtier, 2006*; *Leffler et al., 2012*;

*Lewontin, 1974*; *Romiguier et al., 2014*), we predict *N. meningitidis* to have a much larger $N_e$ than *N. gonorrhoeae*, simply because it infects many more people.

In addition to the influence of hLGT we see the signature of recombination between strains of the same species breaking down LD, since LD decreases with increasing distance between sites. Similar patterns have been previously reported in both *N. meningitidis* (*Budroni et al., 2011*) and *N. gonorrhoeae* (*Arnold et al., 2018*) but these studies used different LD statistics and so it is hard to determine what the comparative patterns are. The patterns are similar in the two species, but they are consistent with a difference in $N_e$ since the decay in LD is faster in *N. meningitidis* and asymptotes at a slightly lower value. In both species the asymptote is above what is expected under free recombination even taking into account sampling error and the fact that $r^2$ cannot be negative (see above). The asymptote might be above this level for two reasons. First, there might be a balance between drift and recombination. In a gene conversion model of recombination, a non-zero asymptote is expected because once sites are further apart than the gene conversion tract length, then increasing distance does not increase the rate of recombination. The asymptote is then determined by a balance between drift increasing LD, and recombination breaking it down. The second explanation is that there is population sub-structure in both species. It has been argued, based on the phylogeny of strains that there is substructure in *N. meningitidis* (*Budroni et al., 2011*; *Kong et al., 2013*) and *N. gonorrhoeae* (*De Silva et al., 2016*; *Ezewudo et al., 2015*; *Grad et al., 2016*; *Lee et al., 2018*). In *N. meningitidis* it has been suggested that this structure arises because different sets of strains have different restriction modification systems (*Budroni et al., 2011*). However, the correspondence between clades of strains and these systems is not clear cut (*Kong et al., 2013*).

We find as others have found in some other species, that diversity varies across the genome in *N. meningitidis* and *N. gonorrhoeae*, and that this variation affects both synonymous and non-synonymous sites. This is in large part driven by hLGT; regions of the genome with high rates of hLGT have high diversity. However, when we focus on the part of the genome that is inferred not to have undergone hLGT we find that levels of non-synonymous and synonymous diversity are correlated, but in a manner which demonstrates that $\pi_N/\pi_S$ declines with increasing $\pi_S$. A similar pattern has been observed within the genomes of various eukaryotes (*Castellano, James & Eyre-Walker, 2018*; *Gossmann, Woolfit & Eyre-Walker, 2011*; *Murray et al., 2017*) as well as between eukaryotic species (*Chen, Glemin & Lascoux, 2017*; *Galtier, 2016*; *James, Castellano & Eyre-Walker, 2017*). This pattern is consistent with an influence of linked selection on the genome—regions of the genome with high levels of linked selection have low $\pi_S$, but relatively high levels of $\pi_N$. Linked selection can influence diversity in two ways. First, both background selection and genetic hitch-hiking can reduce the effective population size of a genomic region. Second, hitch-hiking can lead to non-equilibrium dynamics which can affect the relative levels of selected and neutral diversity; after a hitch-hiking event deleterious genetic diversity will return to its equilibrium value faster than neutral diversity (*Brandvain & Wright, 2016*; *Do et al., 2015*; *Gordo & Dionisio, 2005*).

*N. meningitidis* and *N. gonorrhoeae* occupy distinct niches and one might presume that they have undergone adaptive evolution. Such adaptation might have been

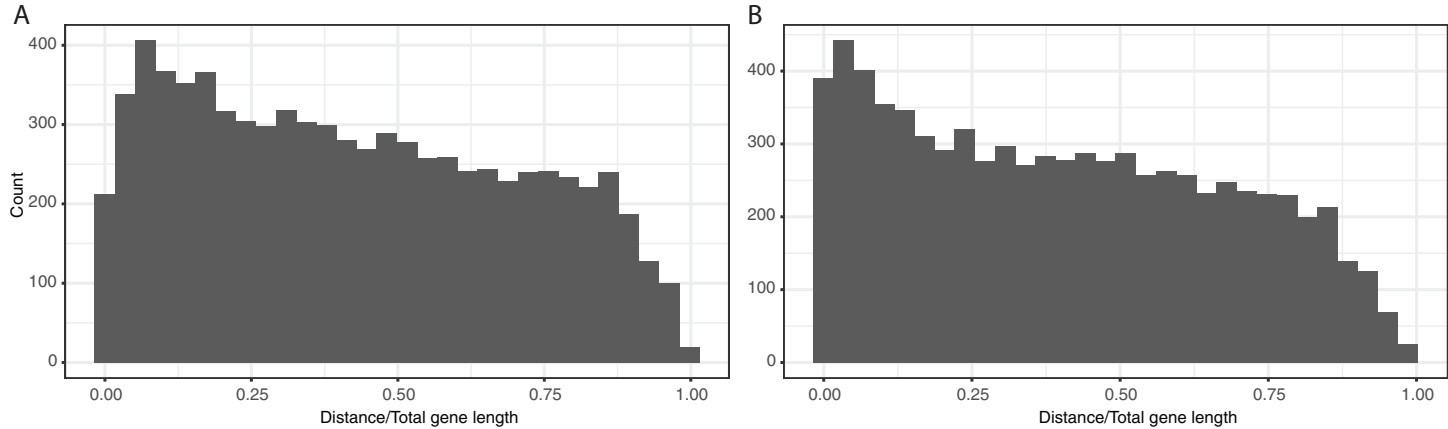

**Figure 5** **hLGT tracts at the start and end of genes.** The number of sequences inferred to be due to hLGT in both species as a function of the distance from the (A) start and (B) end of genes, where the distance was the proportion of the gene length from the start and end.

achieved through the acquisition of new genes, and/or adaptation in their core genomes. We have tested for adaptive evolution in the core genome using two approaches—a McDonald–Kreitman test in which numbers of non-synonymous and synonymous substitutions are compared to numbers of non-synonymous and synonymous polymorphisms (*Eyre-Walker, 2006*; *McDonald & Kreitman, 1991*). We find significant evidence of adaptation when we compare the substitution data to the polymorphism data of *N. meningitidis*, but no evidence if we use the polymorphism data of *N. gonorrhoeae*. These observations are consistent with a decrease in the $N_e$ of *N. gonorrhoeae* or an increase in *N. meningitidis* (*Eyre-Walker, 2002*). The difference in $N_e$ is consistent with the observation of higher diversity in *N. meningitidis*, lower $\pi_N/\pi_S$, more rapid decay in LD and the lower asymptote in LD. However, it is difficult to resolve whether *N. gonorrhoeae* has undergone population size contraction or *N. meningitidis* population size expansion in the past. Finally, it is tempting to estimate the fraction of substitutions fixed by adaptive evolution as $1 - 1/FI$—see (*Eyre-Walker, 2006*). However, the simultaneous introduction of multiple mutations by hLGT makes this estimate biased.

A central assumption in our analysis is that ClonalFrameML (*Didelot & Wilson, 2015*) has correctly identified regions of the genome that have undergone hLGT. The method identifies the presence of hLGT from a clustering of mutations along an inferred clonal phylogeny; a sudden burst of mutations along a branch in the phylogeny, that are spatially clustered together in the genome are inferred to be due to hLGT. It will therefore be difficult for the method to detect hLGT with relatively similar or short sequences. Furthermore, because we have used a concatenation of protein coding sequences in our ClonalframeML analysis it may be difficult to detect hLGT at the start and end of genes, because we will not have the flanking sequences which provide additional support for hLGT. To investigate whether this latter effect is important, we plotted the number of inferred hLGT events as a function of the distance from the start or end of genes. We found that events are inferred slightly less often at the start/end of genes, but the effect is not large (Fig. 5).

The fact that ClonalFrameML has probably missed some hLGT events suggests that we may have underestimated the input of variation from hLGT in both species—i.e. we have underestimated $r/m$. However, an inability to correctly detect all hLGT events is unlikely to explain the differences in the relative contribution of hLGT and mutation in the two species, since both species have been treated identically. An inability to detect hLGT may, however, explain why $\pi_N$ and $\pi_S$ are correlated even in the parts of the genome with no apparent hLGT and hence there may be little or no variation in $N_e$ across the genomes of *N. meningitidis* and *N. gonorrhoeae*; there is an expectation that $\pi_N/\pi_S$ is likely to be lower amongst hLGT fragments because the polymorphisms will be dominated by mutations that are fixed between species. Furthermore, it is possible that all the variation that is shared between *N. meningitidis* and *N. gonorrhoeae* is a consequence of hLGT and we have not been able to identify all hLGT events.

## CONCLUSIONS

We have investigated the diversity in *N. meningitidis* and *N. gonorrhoeae*, and shown that *N. meningitidis* is more diverse then *N. gonorrhoeae*. Both species have acquired most of their variation through hLGT. *N. meningitidis* appears to have higher diversity in part due to it's higher effective population size. In both species LD decays relatively slowly as a function of the distance between sites and there is some evidence of adaptive evolution in the core genome of the two species.

### Funding
The project was funded by an Erasmus grant to allow the student Lucile Vigue to visit the lab of Adam Eyre-Walker. There was no other funding. The funders had no role in study design, data collection and analysis, decision to publish, or preparation of the manuscript.

### Grant Disclosures
The following grant information was disclosed by the authors:
Erasmus grant to allow the student Lucile Vigue to visit the lab of Adam Eyre-Walker.

### Competing Interests
The authors declare that they have no competing interests.

### Author Contributions

- Lucile Vigué conceived and designed the experiments, analysed the data, contributed reagents/materials/analysis tools, prepared figures and/or tables, authored or reviewed drafts of the paper, approved the final draft.
- Adam Eyre-Walker conceived and designed the experiments, authored or reviewed drafts of the paper, approved the final draft.

## Data Availability

The sequences are available from the NCBI genome repository:

*N. gonorrhoeae*: NC_011035, NC_022240, NZ_CP012026, NZ_CP012027, NZ_CP012028, NZ_CP016015, NZ_CP016016, NZ_CP016017, NZ_CP020415, NZ_CP020418, NZ_CP020419, NZ_LT906437, NZ_LT906440, NZ_LT906472.

*N. meningitidis*: MC58, NC_003116, NC_008767, NC_010120, NC_013016, NC_017501, NC_017505, NC_017512, NC_017513, NC_017514, NC_017515, NC_017516, NC_017517, NC_017518, NZ_CP007524, NZ_CP009418, NZ_CP009419, NZ_CP009420, NZ_CP009421, NZ_CP016684.

Codes written to analyse the sequences are available as Code S1.

## Supplemental Information

Supplemental information for this article can be found online at http://dx.doi.org/10.7717/peerj.7216#supplemental-information.

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
