# Peer review of "The comparative population genetics of Neisseria meningitidis and Neisseria gonorrhoeae"

_PeerJ, doi:10.7717/peerj.7216_

## Round 0.1 · original submission · Major Revisions

As you will see, the reviewers highlighted some important contributions of your manuscript. However, they have also raised some criticisms that should be addressed before the manuscript can be accepted.

Reviewer 1 ·

Basic reporting

no comment

Experimental design

no comment

Validity of the findings

I really enjoyed reading this paper and was very happy to see these questions asked for bacteria, especially for species I find very interesting.

I have a couple of concerns, and I apologize if any of these concerns are a result of something going completely over my head.

What first caught my eye was how r^2 decays suspiciously fast with the estimated genetic distance between SNPs. This is also likely why the estimated tract lengths in Table 2 are suspiciously small, since estimating bacterial recombination parameters depends on the rate at which r^2 decays in addition to the r^2 value it asymptotes to. My prior to what Neisseria data may look like include Budroni et al. 2011 (PNAS) for N. meningitidis and Arnold et al. 2018 (Genetics) for N. gonorrhoeae. There’s nothing wrong with proving previous studies wrong, but a key methodological difference is that the previous authors either used whole-genome alignments with Mauve (Budroni et al 2011) or also found core genes like you do, but kept track of their positions using a known, contiguous reference genome (Arnold et al 2018). Alternatively, your analyses are done on alignments generated from concatenating core genes (Concatenated how, by the way? Did you attempt to preserve synteny or concatenate them in random orders?). Thus, distances between SNPs may be systematically underestimated if the accessory genes present between core genes are not accounted for. If there is a recent insertion of a rare accessory gene in-between two core genes, this is likely not an issue for you, since most of the evolutionary history of the flanking core genes is spent adjacent to one another just as you have put them in your concatenated alignments. However, there could be intermediate- and high-frequency accessory genes that could cause problems here. These insertions between your core genes could also be very large, making SNPs that are adjacent to one another in your concatenated alignment have linkage dynamics that resemble the genome-wide background, which would make r^2 decay overly quickly with distance.

Inaccurate estimates of distances between SNPs could also affect your ability to detect linked selection, and perhaps why so little of it was detected in N. gonorrhoeae. It may also make patterns even stronger for N. meningitidis.

Given the low sample sizes of genomes here, it could be feasible to generate alignments from progressiveMauve that should preserve synteny. The pangenome tool “Roary” may also make some attempt to keep synteny in the alignments it outputs, but I’m not entirely sure how reliable they are. In any case, I have a hunch this could be an issue, but happy to be proven wrong.

Moreover, while it is acknowledged that clonalFrameML may not be able to detect all interspecies admixture events (since detection of these events is biased towards either longer recombination tracts and/or ones that are from more diverged donor species), I’m not exactly sure how to feel about the analyses in Figure 2+3. For instance, is there any way to get an idea of how much interspecies admixture made it past the clonalFrameML detection step? Your Figure 5 shows that your use of concatenated alignments may have made it more problematic by artificially truncating recombination segments, but even with perfect alignments this could still be a potential issue. Can only linked selection produce negative correlation between piN/piS vs piS, such that even if there were interspecies admixture, this is still evidence? While interspecies admixture can certainly explain the correlation between piN and piS, can it also contribute to the linked selection signal? It seems this could be the case if the donor population was larger and had more efficient purifying selection, making admixture tract lengths introduce more synonymous than nonsynonymous polymorphisms. Nonetheless, I just don’t have the best intuition for this so commenting in the discussion how much this could drive results would be helpful. I understand detecting *all* interspecies recombination is a very difficult task and on ongoing challenge for the field.

Lastly, these two species could have different population structures, resulting in N. meningitidis having larger Ne. To me, this is never really presented as a serious alternative hypothesis in the paper, and all analyses are done basically treating both species as single populations. This is probably valid for N. gonorrhoeae, but may not be for N. meningitidis? In particular, I’m thinking of Smith et al. 1993. I’m not suggesting that a thorough analysis of population structure be done, but it would at least be useful to mention. They do both have values of Tajima’s D close to zero, but it’s hard to interpret if this means no structure or if its from a balance of factors that create positive and negative values of D that, genome-wide, average out to zero (e.g. structure with purifying selection). However, perhaps structure still does not drive lower piN/piS in N. meningitidis, suggesting it truly has larger Ne even if some structure is present.


Minor comments:

For the paragraph starting on line 133, could you please provide more motivation behind why you used different sets of synonymous polymorphisms to ranks gene and to get averages across a group? I kind of see what you did there but am left wondering why you did it.

Line 198 – I would personally appreciate if it could be emphasized that this is a difference in the population mutation rate (2Nu), not a difference in the physical mutation rate (u). This, however, can be argued as a difference in style.

Figure 1 – the two colors are not labeled

Paragraph starting on line 235 – It’s mentioned later in the discussion that a simple model of gene conversion (or bacterial homologous recombination) can also cause linkage metrics to asymptote to non-zero values (i.e. your drift explanation), since recombination doesn’t linearly increase with distance between SNPs such that it’s expected even in the absence of population structure. I think this should also get mentioned here, since it was on my mind until it was eventually mentioned in the discussion.

Line 279 – after you give the slopes for Ng and Nm, there are numbers in parentheses that follow the slopes. What are these numbers? P-values?

Paragraph starting on line 299 – For this analysis do you remove genes that have experienced interspecies admixture? Wouldn’t this deflate levels of divergence? Although why it would tend to deflate dN more than dS (making dN/dS < piN/piS) is not exactly clear to me, and perhaps it shoudn’t. Nonetheless, it isn’t immediately clear to me how to interpret these results in the presence of interspecies admixture.

Line 341 – g in “NG” should be lowercase

Line 367 – or differences in population structure

Line 434 – should “refs” be populated with numbers referring to references?

Reviewer 2 ·

Basic reporting

The manuscript reads like a verbal presentation and not a scientific paper, the wording is ambiguous and vague. For example, throughout the manuscript statements such as: “We investigate…” should be replaced with a statement of fact such as “X was investigated…”; “We detected…” should read “Using method Y [thing] was detected”; “The two species share a modest amount of variation…” what does ‘modest’ mean?

The authors do not appear to have done a through literature review on the population structure of N. menigitidis (Nm) or N. gonorrhoeae (Ng), there is no discussion or mention of lineages, typing, population structure or implications of these in a comparative analysis; see the 2011 ASM - Clinical Microbiology Review by Unemo and Dillion, 2014 ESCMID - Clinical Microbiology and Infection by Read, 2016 Future Microbiology by Watkins, Maiden, Gupta.

There are four references cited in the manuscript that do not appear in the reference section: Ohta 1972, 1977, 1992, and Welch 2008.

The results section contains information that is better suited to the introduction or discussion.

The genomes used were not referenced properly, many have been published.

Figure 4 is not referenced in the manuscript. All tables and figures are lacking full descriptive information and informative labels.

Experimental design

The genomic comparative analysis of the two pathogenic Neisseria species is useful in understanding their relationship within the genus and the techniques used in this manuscript are suitable for this type of analysis. However, the manuscript does not present what problem the analysis is meant to address, what the gap in knowledge is, or what the hypothesis is regarding the problem or gap.

Validity of the findings

The manuscript does not contain any information on how the sub-set of genomes 'from Genbank' were chosen or how the quality of their assembly was assessed. The NCBI (Genbank) lists >90 Nm and >20 Ng genomes filed as ‘complete’ however some of the genomes chosen for this analysis are actually poor assemblies. How was the data set assessed?

No conclusions were presented; what is the added value of this analysis to the field?

Additional comments

The analysis confuses meningococcal colonization (carriage) with invasive infection (disease), for example: Nm asymptomatic carriage rates are presented but the analysis uses Nm invasive disease genomes, which have a different prevalence. The analysis should be redone using a more appropriate data set or correct prevalence rates.

There are inconsistencies through out the paper; for example: the introduction states a 8-25% rate of carriage, yet in the discussion (citing the same 2004 and 2005 papers) states the rate is 5-20%. Additionally, there are more current publications of infection and carriage rates; for example see the WHO 2018 Meningococcal meningitis report and the CDC Meningococcal Disease web page.

Reviewer 3 ·

Basic reporting

The overall structure of the manuscript is fine, with all results presented in an adequate order, but the style of writing, lack of properly formatted equations and the style of referring to previous works and techniques, makes it very hard to read. This readability issue may have affected the clarity in the exposition of novel results.

A main issue in this article is that it is not entirely clear how the reported results will contribute to the literature in Neisseria or the knowledge of processes affecting diversity of bacteria. For the most part, results are presented as an exploration of some population genetic measures and comparison of these measures between two species of Neisseria. The authors might want to discuss more throughly how their results in Neisseria compare to those from other bacteria and how they lead to novel findings about the strength of the evolutionary processes acting on these bacteria.

Experimental design

The experimental design of this work, per se, does not require deep revisions, unless the authors could figure out a way to either independently estimate the effective population size for the two species under analysis or to separate its effect from those of selection.

Since the above is hard, I think this work covers all that can be extracted from this data but, as mentioned above, should undergo extensive review for better clarity. Before such revision, accessing the relevance of this work is hard.

Validity of the findings

Although well performed, It is not clear whether these results actually add something, even as replication. Other works in the same field either (i) introduced novel analysis techniques and models or (ii) compared and inferred general evolutionary processes acting on the organisms studied. I cannot identify such features in the current manuscript.

Additional comments

I believe this work will be publishable after extensive review of its writing style, as stated above.
Since these revisions should affect only style and presentation, I consider it to be minor revisions.

More detailed corrections, specific questions and suggestions that the authors should address were added as comments in the attached PDF.

Annotated reviews are not available for download in order to protect the identity of reviewers who chose to remain anonymous.

---

## Round 0.2 · Minor Revisions

The authors have made most of the corrections suggested by the editors, except for one of suggestion to improve scientific writing standards.

Reviewer 1 ·

Basic reporting

I appreciate the authors responses to my previous comments as well as addressing concerns of the other reviewers. My main concern was about the discrepancy between the LD decay curves and how they compare with previous findings, although I agree this completely depends on the samples used. While I think the science is sound, I do think the writing could be improved in numerous places to make it more professional. There are many instances of syntax errors and fewer instances of sentences that are not clear. I give some examples below.


Line 93 – strange use of comma, also based on how the previous paragraphs are set up, while this paragraph starts off detailing the “second question”, I’m not quite sure what the first was.

Line 107 – confusing sentence structure of sentence starting on this line, and “may be” should probably be “maybe” based on the context.

Line 119 – random, not “randomly”

The names of the strains used and the previous studies they came from should be put into a table, not listed in line since there are quite many.

Line 143 – there’s a notification of an invalid citation, this should be fixed

Line 160 – sentence starting on this line is awkward, can probably remove “, in the strains affected” since this is implied from the first part

Line 170 – sentence on this line not correct, which Microsoft Word also marked with a green squiggly line underneath. I suggest going back through the manuscript and checking all places Word suggests changing sentence structure, as I agree with it on some occasions.

Line 174 – I think there should be a comma after “positive”

Line 176 – it is never mentioned how these bi-allelic loci are simulated, what program? And what frequencies? Later on, on Line 179, it’s mentioned that the expected value of r^2 is independent of allele frequencies which does not make sense to me, as VanLiere and Rosenberg (2008; Theoretical Population Biology) show this can’t be true. The maximum value of r^2 depends on allele frequencies. So, this must either be explained further, but I suppose you could also delete this sentence because it’s also not clear to me what this information is used for in the manuscript.

Line 213 – sentence starting on this line should be restructured

Line 236 – “they introduced” should be changed to “they are introduced”

Line 248 – “recombination initiates” isn’t followed by a noun. What does it initiate?

Line 340 – should be “spread of an advantageous mutation”

Line 357 – again, awkward sentence here for same reason as above, where “may be” should probably be “maybe”

Line 392 – the e in Ne is not subscripted as before, please go through and change this at other locations if necessary too.

Line 472 – Ng should be followed by a comma, which Word identified with a green squiggly line.

Line 493 – I think you have the citations wrong here, I think Budroni et al studied Nm, not Ng.

Line 503 – Again, this sentence is improperly formatted and Word flagged it as well.

Line 518 – It’s not clear to me what the difference between “genomes of various eukaryotes” and “between various eukaryotic species” is, please clarify

Line 566 – missing comma

Line 579 – I do not understand what you mean here, you use “differences” twice, and these referring to distinct things? Unclear.

Experimental design

I have no comments on the experimental design.

Validity of the findings

I have no comments on the validity of the findings, just how they are presented (see above).

Reviewer 2 ·

Basic reporting

Over all the writing style has not been amended to follow scientific writing standards and still affects the clarity and potential impact of the manuscript.

Line 20, 124 - numerals under 10 should be spelled out
Line 26 - ‘LD’ should be defined
Line 53 - last sentence is missing a full stop
Line 74 - ‘SNPs’ should be defined
Line 100-102 - this sentence needs to be written more clearly, or expalined better
Line 115 - change ‘randomly’ to ‘random’
Line 139 - remove ‘(!!!INVALID CITATION!!! ;)’
Line 161, 190, 192-3, 195, 197, 219, ect. - replace ‘is’ with ‘was’
Line 190-193, 219-229,241-243, 262-266, etc. - this is not a result, it is a discussion point
Line 206-211 - this is not a result, and should be in the methods section
Line 257 - please check the validity of the percentage ‘-15.5%’
Line 346-354 - correct abbreviation inconsistency of fixation index
Line 349, 353 - please indicate what the numbers in parenthesis refer to

Experimental design

n/c

Validity of the findings

Each Nm lineage has it's own distinct molecular epidemiology, the manuscript should address the impact of using multiple Nm lineages on the results. For example:
Line 22 - while recombination is present in Nm, the diversity is just as likely to be a result of the multiple lineages used in the study
Line 200-201 - please clarify: does ‘polymorphism’ refer to the gene diversity in both species, or that the two species sharing common alleles of certain genes?
Line 418-419 - while the Ng strains can be considered ‘very closely related’ the Nm strains are from different lineages and are not ‘very closely related’ to each other
- The Nm strains used in this study represent multiple distinct evolutionary lineages, each with a different fitness potential and epidemiology which can be highly invasive or asymptomatically cleared by the host. Conversely, Ng is invasive, pathogenic, highly clonal, and does not form distinct lineages.
Line 440 - please reference the comment that the Ng infection rate is lower than Nm; the comment does not represent official reported figures from the ECDC and CDC
- Nm in the USA was 0.12 cases per 100,000 in 2016 (www.cdc.gov), and in the EU/EEA 0.6 cases per 100,000 in 2016 (ecdc.europa.eu)
- Ng in the USA was 171.9 cases per 100,000 in 2017 (www.cdc.gov); while in 2017 in Europe rates varied from 1 to 75 cases per 100,000 population depending on the country and in total 89,000 confirmed Ng diagnoses in 2017 (ecdc.europa.eu)

Reviewer 3 ·

Basic reporting

I believe the article now describes, with enough clarity, its results and conclusions.
The goals of comparing population genetics parameters between Nm and Ng is achieved and, although if it seems to me a preliminary incursion into this topic, I believe it is worth publishing.

Experimental design

no comment

Validity of the findings

no comment

---

## Round 0.3 · accepted · Accept

The authors have addressed the reviewers' concerns satisfactorily. Hence, I recommend the publication of this manuscript in PeerJ.